# An after-action review of COVID-19 cases and mitigation measures at US Mission India, March 2020-July 2021

Jaspreet Singh[1]*, Rajesh Yadav[2], Samantha Robinson[1], Mark Vanelli[1], Melissa Nyendak[2], Meghna Desai[2]

**1** Health Unit, U.S. Embassy, New Delhi, India, **2** U.S. Centers for Disease Control and Prevention, U.S. Embassy, New Delhi, India

* dr.singh.jaspreet@outlook.com

## Abstract

Between March 2020-June 2021, over 30 million COVID-19 cases were reported in India. We described the COVID-19 response across the US Mission India (US Embassy New Delhi, US Consulates – Mumbai, Hyderabad, Chennai, and Kolkata) to use the learnings for a possible future pandemic. We reviewed COVID-19 mitigation activities at five US Mission India posts from March 2020–July 2021. We also analyzed case investigation and contact tracing data from Health Units (outpatient clinics), including demographics, clinical findings, test results, contact positivity rate, and compared attack rates across the posts during the same period. The US Mission in India, comprising multiple US Government agencies, initiated COVID-19 mitigation in March 2020 with educational sessions, infection prevention training, health assessments, and standard operating procedures. The Health Unit and US CDC India office initiated COVID-19 case investigations and conducted contact tracing. During the study period, 636 COVID-19 cases (72% males), including 48 clusters (size range 2–10 cases), were reported. Overall case fatality rate was 1.5% (10). Of case patients, 82% (523) were Indians, and 18% (113) were Americans. On presentation, 22% (138/625) of cases were asymptomatic. The median time from symptom onset to notification to the Health Unit was three days (Interquartile range 1–5). The Health Unit identified 2,484 contacts with a 25% positivity rate. The attack rates ranged between 10–19%, with the highest at 19% in Delhi, which was lower compared to the estimated attack rate for respective cities but closely resembled the pattern of COVID-19 waves in India. Collaboration between medical providers and public health specialists during the COVID-19 response in US Mission India led to new organizational capabilities in contact tracing, community education, and workflows. These strategies helped reduce morbidity and mortality within the US Mission during the pandemic.

**Data availability statement:** The data includes details related to US Mission India staffing and health outcomes from a secondary analysis of COVID-19 contact tracing efforts. NAME OF THE RESTRICTING INSTITUTION- US EMBASSY, NEW DELHI - EMAIL CONTACT - NDCOVID19CONTTRAC@STATE.GOV - Any specific data access requests will be catered to after approval from the appropriate authorities.

**Funding:** The author(s) received no specific funding for this work.

**Competing interests:** The authors have declared that no competing interests exist.

## Introduction

Coronaviridae are an RNA family of viruses that have the tendency to jump from animals to humans [1]. In the past two decades, three human coronaviruses have emerged, leading to acute respiratory illness and significant morbidity and mortality, most recent SARS-CoV-2 [2]. COVID-19, the disease caused by infection with SARS-CoV-2, was labeled as a pandemic by WHO in March 2020 due to its rapid rate of spread, associated mortality and its complications [3,4].

Between March 2020 and June 2021, over 30 million cases of COVID-19 were diagnosed in India, including two major waves in September-December 2020 and April-May 2021 [5]. During that period, over 600 COVID-19 cases occurred in the United States Diplomatic Mission to India (US Embassy and Consulates in India) amongst American employees and their family members, Indian employees, contractors, and household staff. Throughout this period, as part of its COVID-19 public health response program, US Mission India implemented COVID-19 mitigation measures and case investigation with contact tracing for all cases. US Mission India further offered and administered COVID-19 vaccination to employees upon availability beginning in April 2021.

As part of an after-action-review exercise to inform future response efforts, we described the COVID-19 response across the five US Mission India posts, including a comparison to cases in India overall and in specific consular cities between March 2020 and July 2021.

## Methods

### Ethics statement

We have obtained approvals from the US CDC (Atlanta) and Bureau of Medical Services (Washington, D.C.), US Department of State, for the secondary data analysis of the pre-existing dataset of COVID-19 public health response at US Mission India. This study did not involve any primary data collection; hence consent from the participants or parents in case of dependent children was not required. This review was deemed non-research by the Office of the Associate Director for Science of US CDC, hence a full Institutional Review Board approval not recommended. The dataset was obtained after stripping of all personal identifying information to maintain the confidentiality of the participants.

### Study design

We described the COVID-19 mitigation activities undertaken by US Mission India between March 2020 - July 2021. Additionally, we performed a descriptive secondary analysis of the available data from the COVID-19 contact tracing program of the US Mission India Health Units (outpatient clinics).

### Study site

US Mission India facilities: Embassy New Delhi, Consulate General Mumbai, Consulate General Chennai, Consulate General Hyderabad, and Consulate General Kolkata. US Mission India facilities included 4399 people (57% at Embassy New Delhi).

### COVID-19 mitigation activities

We collected and compiled information about the US Mission's COVID-19 response and mitigation activities during the study period. This included a review of policy and guidance documents and standard operating procedures; execution of trainings, webinars, COVID-19 group meetings, seminars, town halls, and vaccination drives; and development and dissemination of newsletter articles and management notices.

### Analysis of health unit case and contact investigation program

The COVID-19 contact tracing was done by respective Health Units across Mission India cities led by the US Embassy Health Unit, New Delhi. The centralized database was maintained by the US Embassy Health Unit, New Delhi, in a secured Excel sheet and was used for this analysis.

We included eligible participants (COVID-19 cases) from all five sites.

**Inclusion criteria.** All US direct hires (USDH), eligible family members (EFM), locally employed staff (LES), local guard force (LGF), and others (housekeepers, nannies, and drivers working for Americans and contractors) who either tested positive for COVID-19 by rapid antigen test or Reverse Transcriptase Polymerase Chain Reaction (RT-PCR)/ had a clinical diagnosis of COVID-19 while posted or during travel within India, or tested positive/ had a clinical diagnosis of COVID-19 while on a temporary personal or official visit to India during the study period.

**Exclusion criteria.** All USDH, EFM, LES, LGF, and others (housekeepers, nannies, and drivers working for Americans and contractors) who tested positive for COVID-19/ had a clinical diagnosis of COVID-19 while they were traveling outside of India for official or personal reasons.

A COVID-19 cluster was defined as two or more epidemiologically linked cases during the study period. A COVID-19 contact was defined as someone who was less than 6 feet away from an COVID-19 case (laboratory-confirmed or a clinical diagnosis) for a total of 15 minutes or more over a 24-hour period. We reviewed the COVID-19 case data for all five US Mission cities in India from the Ministry of Health and Family Welfare (Government of India) website for comparison with reported US Mission cases in those respective cities.

### Data analysis

We calculated the distribution frequency of selected variables, including nationality, for each of the US Missions in India. Case investigation and contact tracing data included demographic data for both cases and contacts, date of symptom onset, date of contact by the Health Unit case investigation and contact tracing program, number of contacts, duration of isolation and outcome for cases, contact type (i.e., symptomatic or asymptomatic), and test results for contacts. For all COVID-19 cases, descriptive analysis was performed to assess age, sex, frequency of case identification by Mission, contacts per case, positivity rate among contacts by type of contact (i.e., household, office, or social), and comparison of US Mission case trends with overall case trends in India and Consular cities (Ministry of Health and Family Welfare, Government of India). We used the Chi-square test to assess the association between selected categorical variables and calculated Odds Ratio (OR) with a 95% Confidence Interval (C.I). The data were analyzed using Epi Info Software, version 7.2.4.1 (US CDC).

## Results

The US Mission in India reported its first COVID-19 case in March 2020 from New Delhi. Between March 2020 and July 2021, a total of 636 COVID-19 cases, including 48 clusters, were reported from the US Mission – India (Fig 1). About 72% of the cases were males, and the median age was 42 years, with an interquartile range of 35–49 years. Overall, 82% (523) were Indians, and 18% (113) were Americans.

In response to COVID-19, the US Mission in India in collaboration with internal US Mission divisions initiated case investigations and contact tracing; education sessions for mission staff; targeted infection prevention and control training;

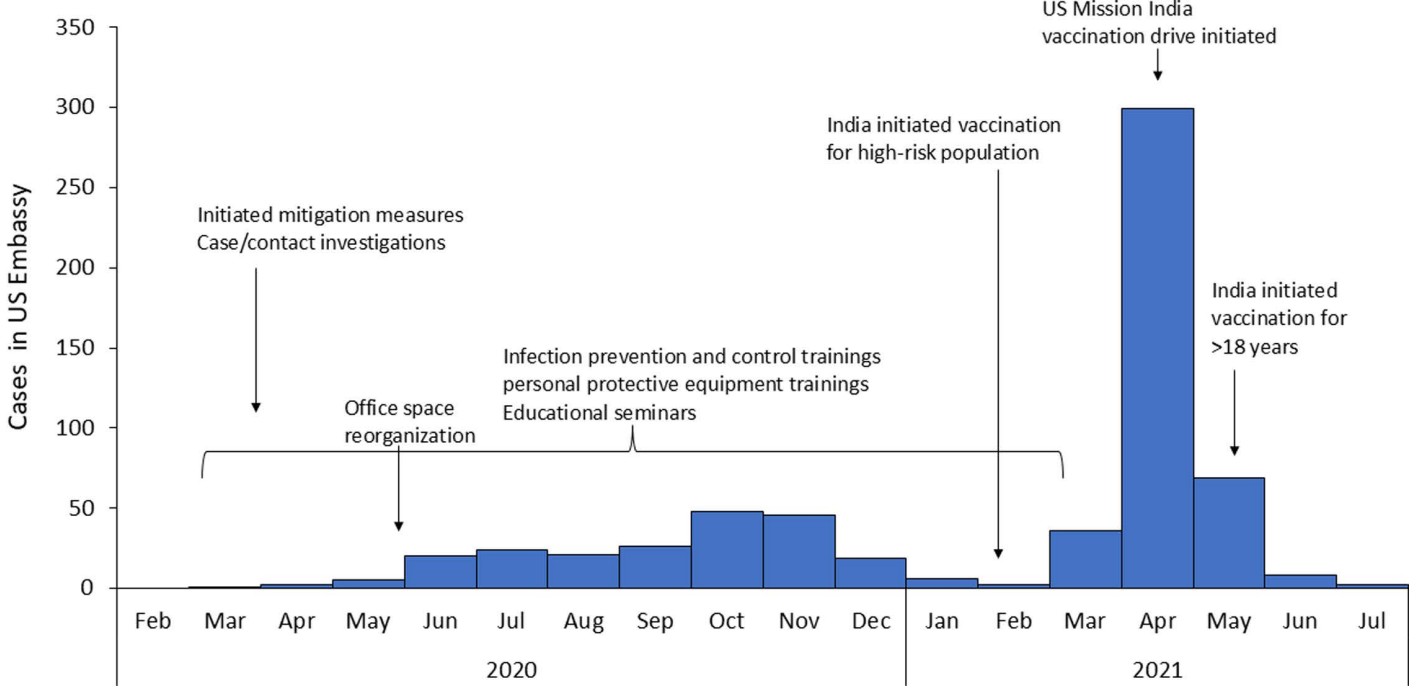

**Fig 1. COVID-19 mitigation activities undertaken at US Mission in India.**

workplace health and safety assessments; development of standard operating procedures; symptom-self-screening; and COVID-19 vaccination. (Fig 1).

## Case investigation and contact tracing

The US Mission Health Unit and US Centers for Disease Control and Prevention (CDC) India office initiated a program for COVID-19 case investigations and contact tracing follow-ups. The case investigation and contact tracing team closely followed COVID-19 guidance in India (Ministry of Health, Government of India and different State Governments) and around the world to implement Indian guidance (the law of the land) in coordination with US guidance (being a US Mission). Where there was divergence, the US Mission followed the stricter of the two to ensure that both Indian and US guidance was followed. Familiarity with guidance in other countries was also important within this diplomatic community as some essential official travel was required even during the lockdowns. We ensured that all cases and contacts completed government-recommended isolation or quarantine and underwent the recommended COVID-19 testing based on exposure and symptoms onset per the guidelines.

## Educational sessions for mission staff

Health education and risk communication through open forum webinars and virtual town halls, which were conducted in both English and Hindi for better outreach. Periodic management notices and newsletter articles were shared with the Mission community, informing them about the sessions. Between May 2020 and July 2021, the US Mission collaborated with the COVID-19 health experts of US Mission India to conduct a weekly "Doctors are In," Webex seminar to address questions from the US Mission community.

### Targeted infection prevention and control training

Sessions included hands-on demonstrations for various sections of the workforce, including supervisors, contract cleaning staff, locally employed staff (LES), motor pool, and cafeteria staff. Other training included personal protective equipment training, cleaning and infection training, social distancing training, and hand washing training.

### Workplace health and safety assessments

Assessments were done for congregate workplace settings, and recommendations were provided for modified layouts to promote physical distancing, improved ventilation, and air circulation, and cleaning and sterilization guidance [6].

### Development of standard operating procedures

US Mission India Health Unit, CDC, and other agencies collaborated and developed guidelines for COVID-19 to maintain the continuity of the Mission's critical activities. Workplace risk reduction guidelines included guidance on mask use, physical distancing, staying at home when ill, increased telework and virtual meetings, and temperature screening and symptom checking. Other guidance topics included monitoring and evaluating COVID-19 cases and contacts, COVID-19 case cluster identification and response, travel-associated guidance for domestic and international travel, and case isolation and contact quarantine support.

### Symptom self-screening

To reduce the risk of COVID-19 transmission in the workplace, the US Embassy Health Unit implemented the strategy of symptom self-screening. All employees were instructed to stay at home and update the Health Unit if they are sick, if anyone in their family (living in the same household) is sick, if they test positive for COVID-19, or if anyone in their family tests positive for COVID-19. Employees required clearance from a medical provider to return to work. Numerous other mitigation measures were added, including temperature screenings, a mask mandate, the installation of hand sanitizers (with foot-operated dispensers) outside every building, and the promotion of telework.

### COVID-19 vaccination

Once the COVID-19 vaccine became available in April 2021, the Health Unit procured m-RNA vaccines (Pfizer and Moderna) from the Bureau of Medical Services, US, and implemented vaccination drives for all Mission staff and community. While conventional COVID-19 vaccine (Covishield and Covaxin) was made available by the Indian government starting February 2021.

**Descriptive analysis of the COVID-19 cases in the US Mission in India.** A total of 10 deaths of the 636 cases (case fatality rate of 1.5%) were recorded among the US Mission, eight of which occurred in 2021. Of these 10 deaths, nine were at the US Embassy in New Delhi, while one was from the Chennai consulate.

The distribution of COVID-19 cases in the US Mission India closely resembled the pattern of COVID-19 cases in the respective cities of India. The peak in cases in 2020 came early in Chennai, Hyderabad, and Mumbai from June-August compared to October-November in Delhi and Kolkata. In April 2021, cases peaked in Delhi, Chennai, Hyderabad, Kolkata, and Mumbai (Fig 2).

We identified and responded to 48 COVID-19 clusters across the US Mission in India. The median size of the cluster was 2 cases (range 2–10 cases), and 80% (38) clusters included ≤3 cases.

In 2020, we observed that the overall attack rate was lower in Americans (3%) compared to Indians (5%) (p-value 0.007). However, at the Kolkata consulate, Americans had the highest attack rate of 16% compared to 4% among Indians (p-value 0.04). In 2021, the attack rate was similar between Americans (10%) and Indians (10%) (p-value 0.9). However, at the Kolkata consulate, Americans had the highest rate of 21% compared to 7% among Indians. During 2020–21, the overall attack rate across all consulates was similar among Americans (13%) compared to Indians (15%) (p-value 0.1) (Table 1).

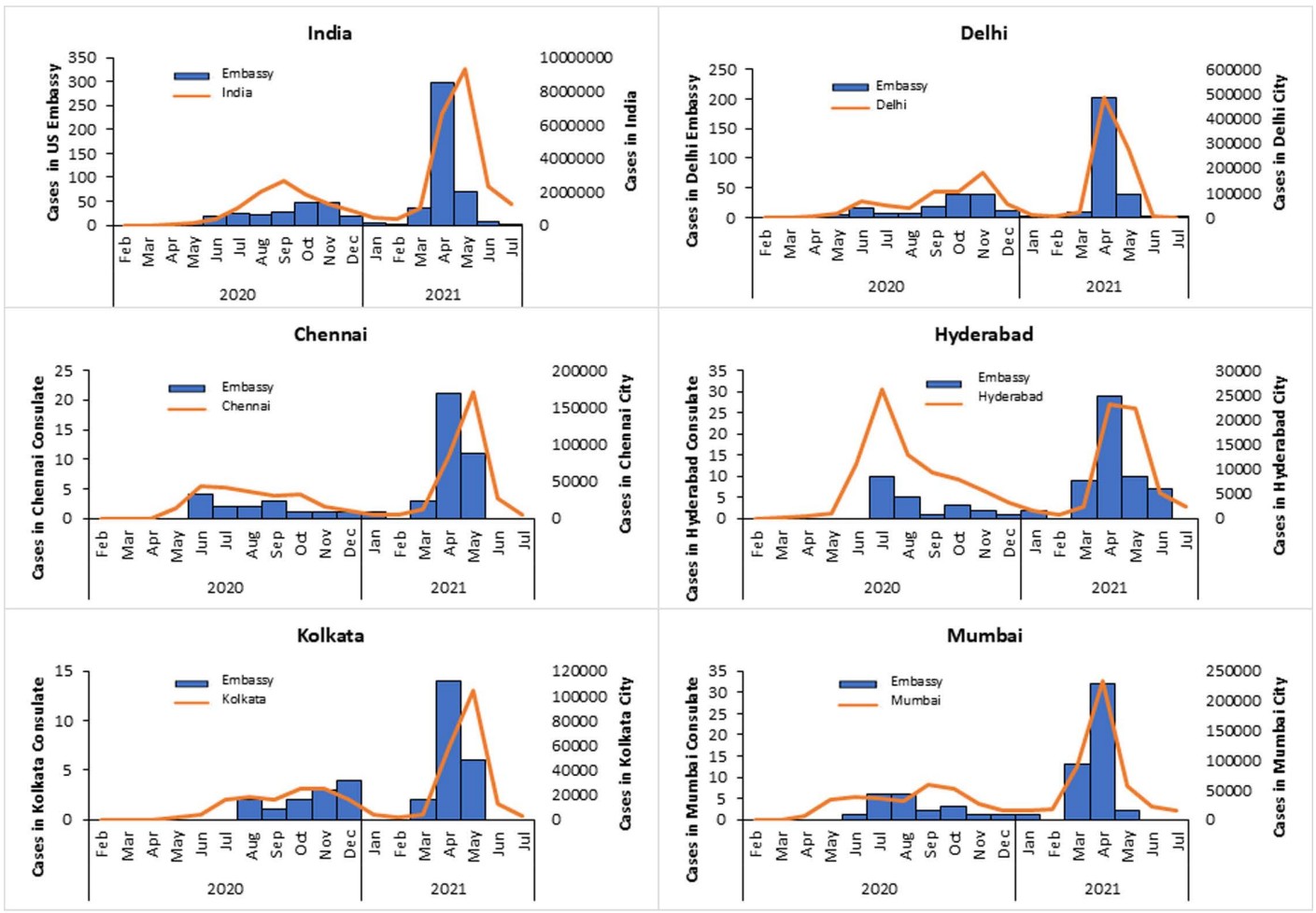

* Two case information were incomplete.

**Fig 2. COVID-19 cases by month in US Mission, India, Mar 2020-July 2021 (n = 634).**

Individuals between the ages of 41–51 years accounted for the highest caseload, while those above the age of 60 years had the highest case-fatality rate (Table 2). At the time of presentation at the Med Unit, about 22% (138/636) of cases were asymptomatic. Among the 498 symptomatic cases, 63% reported fever, followed by body aches (48%), cough (43%), headache (30%), and sore throat (30%) (Fig 3).

Among the 627 cases who reported their smoking status, 27% (167/627) had comorbidities, and 15% (96/627) were either current or past smokers. Among 167 cases with comorbidities, 57% (96/167) reported hypertension, with a 5% (5/96) mortality rate (Table 3). Moreover, 2% (11/627) reported reinfection with COVID-19.

The case fatality rate (CFR) was 3.5% (8/230) among those aged >45 years compared to 0.5% (2/398) among those < 45 years (Odds Ratio [OR] 7.1, CI:1.5-34, p = 0.004). Among cases with both hypertension and diabetes, the CFR was 11% (4/37) compared with 1% (6/391) among those without either comorbidity (OR: 11.9, CI:3.2-44.5, p < 0.0005). The fatality rate reported in US Mission India was 2.5 per 1000 population at risk (Tables 1 and 2). (New Delhi was 3.57 per 1000 population at risk, and Chennai was 2.01 per 1000 population at risk).

**Table 1. COVID-19 Cases and attack rate in US Mission India, Mar 2020-July 2021 (n = 636).**

**2020**

| Consulate | American | n | A.R. % | Indian | n | A.R. % | Total | N | AR % | C.I 95% |
|---|---|---|---|---|---|---|---|---|---|---|
| Chennai | 2 | 92 | 2 | 12 | 405 | 3 | 14 | 497 | 3 | 1.7-4.6 |
| Delhi | 19 | 536 | 4 | 125 | 1984 | 6 | 144 | 2520 | 6 | 4.9-6.7 |
| Hyderabad | 1 | 86 | 1 | 21 | 320 | 7 | 22 | 406 | 5 | 3.6-8.0 |
| Kolkata | 3 | 19 | 16 | 11 | 249 | 4 | 14 | 268 | 5 | 3.1-8.6 |
| Mumbai | 3 | 145 | 2 | 17 | 563 | 3 | 20 | 708 | 3 | 1.8-4.3 |
| Total | 28 | 878 | 3 | 186 | 3521 | 5 | 214 | 4399 | 5 | 4.3-5.5 |

**2021**

| Consulate | American | n | A.R. % | Indian | n | A.R. % | Total | N | AR % | C.I 95% |
|---|---|---|---|---|---|---|---|---|---|---|
| Chennai | 7 | 92 | 8 | 29 | 405 | 7 | 36 | 497 | 7 | 5.2-9.7 |
| Delhi | 49 | 536 | 9 | 210 | 1984 | 11 | 259 | 2520 | 10 | 9.2-11.5 |
| Hyderabad | 8 | 86 | 9 | 49 | 320 | 15 | 57 | 406 | 14 | 11.0-17.8 |
| Kolkata | 4 | 19 | 21 | 18 | 249 | 7 | 22 | 268 | 8 | 5.5-12.1 |
| Mumbai | 17 | 145 | 12 | 31 | 563 | 6 | 48 | 708 | 7 | 5.1-8.8 |
| Total | 85 | 878 | 10 | 337 | 3521 | 10 | 422 | 4399 | 10 | 8.7-10.5 |

**2020-21**

| Consulate | American | n | A.R. % | Indian | n | A.R. % | Total | N | AR % | C.I 95% |
|---|---|---|---|---|---|---|---|---|---|---|
| Chennai | 9 | 92 | 10 | 41 | 405 | 10 | 50 | 497 | 10 | 7.7-13.0 |
| Delhi | 68 | 536 | 13 | 335 | 1984 | 17 | 403 | 2520 | 16 | 14.6-17.5 |
| Hyderabad | 9 | 86 | 10 | 70 | 320 | 22 | 79 | 406 | 19 | 15.9-23.6 |
| Kolkata | 7 | 19 | 37 | 29 | 249 | 12 | 36 | 268 | 13 | 9.8-18.0 |
| Mumbai | 20 | 145 | 14 | 48 | 563 | 9 | 68 | 708 | 10 | 7.6-12.0 |
| Grand Total | 113 | 878 | 13 | 523 | 3521 | 15 | 636 | 4399 | 14 | 13.4-15.5 |

**Table 2. Age group wise cases and mortality rate among COVID-19 cases, US Mission India, March 2020 - July 2021.**

| Age group | Cases | Percent | Mortality | Case fatality rate (%) | Confidence Interval (95%) |
|---|---|---|---|---|---|
| 1 - < 11 | 9 | 1 | 0 | 0 | – |
| 11 - < 21 | 13 | 2 | 0 | 0 | – |
| 21 - < 31 | 78 | 12 | 0 | 0 | – |
| 31 - < 41 | 176 | 28 | 0 | 0 | – |
| 41 - < 51 | 220 | 35 | 3 | 1.4 | 0.5-4 |
| 51 - < 61 | 121 | 19 | 3 | 2.5 | 0.8-7 |
| 61 - < 71 | 11 | 2 | 2 | 18 | 5-47 |
| 81 - < 90 | 1 | 0 | 1 | 100 | 21-100 |
| TOTAL | 629 | 100 | 9 | 1.5 | 0.7-2.6 |

The median time from symptom onset to the notification (in days) to the Health Unit in Mission India was three days (Interquartile range 1–5). The median time from symptom onset to COVID-19 testing was three days (interquartile range 2–15 days). The most frequently reported reason for the delay in notification to the Health Unit in Mission India among cases was the perception of symptoms as fatigue (25%) and delay from the local treating physician in suspecting COVID-19 (21%) (Fig 4).

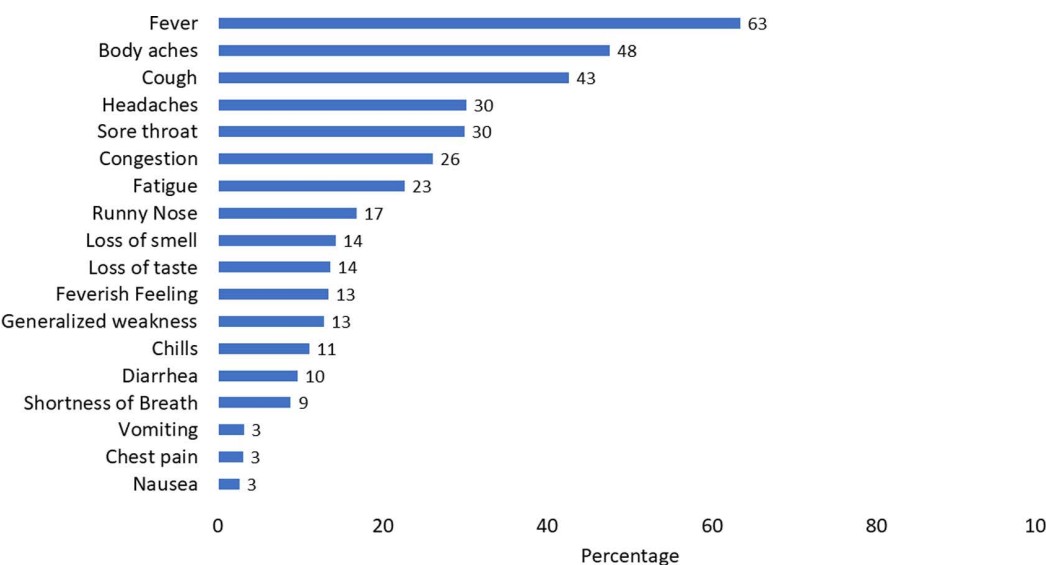

**Fig 3. Symptoms at presentation among COVID-19 cases, US Mission India, March 2020 - July 2021 (n =498).**

**Table 3. Comorbidity among COVID-19 cases, US Mission India, March 2020 - July 2021 (n=167).**

| Comorbidity | Number* | % | Deaths | % |
|---|---|---|---|---|
| Hypertension | 96 | 57 | 5 | 5 |
| Diabetes | 69 | 41 | 4 | 6 |
| Asthma | 50 | 30 | 1 | 2 |
| Heart Disease | 11 | 7 | 1 | 9 |
| Lung disease | 3 | 2 | 0 | 0 |
| Liver disease | 1 | 1 | 0 | 0 |
| Hypertension and diabetes | 37 | 22 | 4 | 11 |
| | | | | |
| Any one comorbidity with age>45 years | 99 | 59 | 6 | 6 |
| Any two comorbidities with age>45 years | 38 | 23 | 5 | 13 |
| Any three comorbidities with age>45 years | 8 | 5 | 1 | 13 |
| Hypertension and diabetes with age>45 years | 25 | 15 | 4 | 16 |
| Hypertension, diabetes, and heart disease with age>45 years | 2 | 1 | 1 | 50 |

*Numbers are not mutually exclusive

There were different tests available for COVID-19 diagnosis in India, namely, Rapid Antigen Test, RT-PCR, CB-NAAT, and Bio-fire, which had varying degrees of sensitivity and specificity. In our analysis, we found that 485 COVID-19 cases were positive on RT-PCR (76%), while 140 COVID-19 cases were positive on Rapid Antigen test (22%), while the rest were clinical diagnoses (2%). Cycle Threshold Scores (C.T. score) for RT-PCR were available for 273 cases with RT-PCR (median was 19 with an interquartile range of 17–21). Overall, 8% (51/619) cases reported travel history within two weeks before the infection. Among the 51 with travel history, 44 reported domestic travel (14 in 2020 and 30 in 2021), and 11 reported international travel (1 in 2020 and 10 in 2021).

Case Management: Two-third of the cases received Vitamin C, Zinc, and Vitamin D as part of the COVID-19 treatment, and half received Ivermectin and Azithromycin. (Fig 5) Hospitalization was 7% (45/627); 44 Indian and 1 American. Six

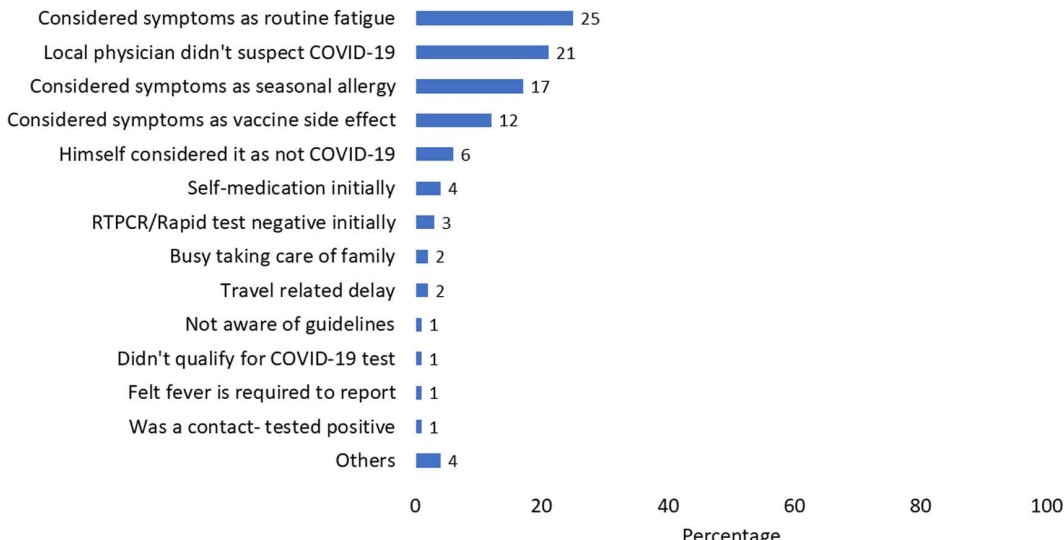

**Fig 4. Perception among COVID-19 cases for delay in notification COVID-19 cases, at US Mission India, March 2020 - July 2021 (n =353).**

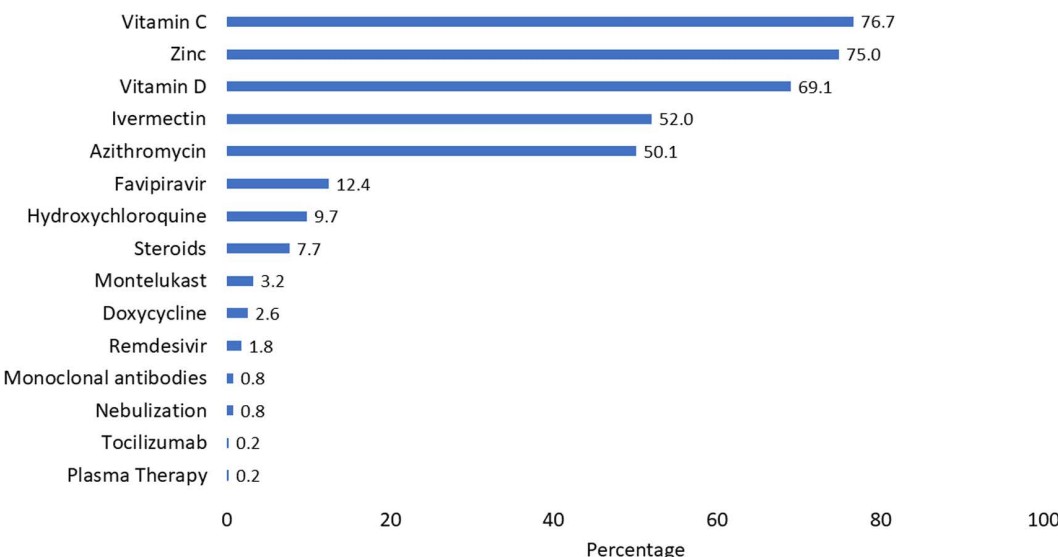

**Fig 5. Treatment history among COVID-19 cases US Mission India, March 2020 - July 2021 (n =627).**

percent (20/326) who received Ivermectin as part of COVID-19 treatment got hospitalized compared to 8% (25/310) who did not receive Ivermectin showing no statistically significant difference (Odds Ratio 0.7, C.I:0.4-1.37, p 0.8). Indians followed the Government of India's Treatment guidance (under the care of their family physicians), while Americans followed US guidelines (under the care of respective Health Units, US Mission India).

As per the prevalent guidance of the Ministry of Health and Family Welfare, Government of India guidelines, all 636 reported cases were isolated, and contact tracing was done for each reported case (found 2,484 contacts). Close contacts

Table 4. Contact positivity among COVID-19 cases US Mission India, March 2020 - July 2021 (n=627).

| Consulates | Office Contact | | | Household Contact | | | Social Contact | | | Family Contact | | | Total | | |
|---|---|---|---|---|---|---|---|---|---|---|---|---|---|---|---|
| | n | +* | % | n | + | % | n | + | % | n | + | % | n | | % |
| Chennai | 29 | 1 | 3 | 4 | 0 | 0 | 0 | 0 | 0 | 100 | 33 | 33 | 133 | 34 | 26 |
| Delhi | 390 | 37 | 9 | 103 | 15 | 15 | 56 | 5 | 9 | 1173 | 394 | 34 | 1722 | 451 | 26 |
| Hyderabad | 136 | 19 | 14 | 11 | 0 | 0 | 13 | 0 | 0 | 169 | 48 | 28 | 329 | 67 | 20 |
| Kolkata | 23 | 1 | 4 | 10 | 7 | 70 | 1 | 0 | 0 | 88 | 24 | 27 | 122 | 32 | 26 |
| Mumbai | 91 | 2 | 2 | 11 | 5 | 45 | 0 | 0 | 0 | 76 | 33 | 43 | 178 | 40 | 22 |
| Overall | 669 | 60 | 9 | 139 | 27 | 19 | 70 | 5 | 7 | 1606 | 532 | 532 | 2484 | 624 | 25 |

A contact was defined as anyone with exposures to a COVID-19 case, from 2 days before to 14 days after the case's onset of illness: being within 1 meter for >15 minutes or direct physical contact or providing direct care without using proper personal protective equipment.

*+=COVID-19 positive

were recommended quarantine, monitored for COVID-19 symptoms, and tested on the fifth day following exposure. The overall contact positivity rate was 25% (range 20–26% among all consulates). Office contact positivity rate was highest in Hyderabad (14%), household contact positivity rate highest in Kolkata (70%), social contact positivity rate highest in Delhi (9%), and family contact positivity rate was highest in Mumbai 43% (Table 4).

Among the 530 who reported their vaccine history through July 2021, 74% had taken Moderna, 10% Covishield, and the rest had other vaccines (Covaxin and Sputnik) available in India.

## Discussion

COVID-19 mitigation strategies were well planned and implemented in collaboration with multiple agencies in the US Mission India. Although the case investigation and contact tracing program led by the Health Unit New Delhi was labor intensive, it helped in preventing the transmission of COVID-19 and its large clusters in the US Mission India. Cases peaked during the waves in each city, which was as expected, but the Mission numbers stayed relatively low [7–9].

Similar to the global experience, one-fifth of the cases at presentation were asymptomatic in our analysis; in previously published studies, this percentage was found to be anywhere between 13% and 31% [10–12].Most deaths reported within the US Mission occurred in the US Embassy New Delhi, but that may be attributed to the large size of the US Embassy in New Delhi compared to the other US Consulates in India. These findings were observed despite being a unique study population (US Mission India staff), who were required to travel within and outside India during the study period.

Our analysis shows there was an eleven times greater chance of mortality amongst individuals aged >45 with co-morbidities such as hypertension and diabetes. As seen in other studies, the presence of chronic conditions such as coronary heart disease, diabetes, and hypertension increased the risk of severe complications of COVID-19 disease [13–17], like our analysis. Other possible reasons worth noting could be the challenges faced at the peak of the second COVID-19 wave in New Delhi, including bed and oxygen crisis [18].

Our analysis found that one of four contacts became COVID-19 positive, which was less compared to other published studies [19]. Given the high positivity rate among contacts, early detection and isolation was the key, as asymptomatic contacts could transmit COVID-19 to others [20].

Perceptions amongst positive cases for delayed notification to the Health Unit included fatigue as a likely symptom of COVID-19, which was more likely to be in individuals requiring physical labor. The second most common reason reported for a delayed notification was the hesitancy of outside physicians to order COVID-19 tests before the appearance of fever, which was not in compliance with the Government of India guidelines. Our analysis showed that fever was present in 63% of the COVID-19 positive patients, similar to other reported studies [21,22].

Although COVID-19 clinical outcomes in diplomatic missions have been described elsewhere [23], our analysis uniquely describes COVID-19 trajectory, outcomes, and mitigation measures, which, to our knowledge, have not been presented amongst any other diplomatic missions.

We performed a secondary analysis on existing data primarily collected for non-systematic COVID-19 contact tracing, with the primary purpose of informing clinical care. As such, some variables of interest were missing, including symptom duration, vaccination details, genomic investigation and analysis. This limited our ability to assess the full clinical spectrum and compare circulating variants within the US Mission community to those circulating in the Consular cities.

US Mission India responded to the COVID-19 pandemic by collaborating with multiple agencies and implementing strategies that likely helped prevent the transmission of COVID-19 and large COVID-19 clusters in the US Mission India. We created pandemic mitigation capabilities that did not previously exist by working across organizational silos, bringing together medical providers, CDC epidemiologists, and multidisciplinary teams to combat COVID-19. We used real-time data, coordinated closely with the host country with ever-changing country-specific guidance and new scientific evidence, and accelerated vaccine distribution once it became available. This innovative approach, flexibility, and coordination helped us keep infection COVID-19 clusters small and minimize deaths. Lessons learned from this pandemic response can guide future responses to pandemics in similar populations.

## Acknowledgments

We acknowledge and thank all the US Mission India Staff who contributed to COVID-19 control and mitigation response.

## Disclaimer

The views expressed in this article are authors' own and not necessarily those of the U.S. Government or US Department of State, Bureau of Medical Services or US Centers for Disease Control and Prevention.

## Author contributions

**Conceptualization:** Jaspreet Singh, Rajesh Yadav, Samantha Robinson, Mark Vanelli, Melissa Nyendak, Meghna Desai.

**Formal analysis:** Jaspreet Singh, Rajesh Yadav.

**Investigation:** Jaspreet Singh.

**Methodology:** Jaspreet Singh, Rajesh Yadav.

**Project administration:** Jaspreet Singh, Rajesh Yadav, Samantha Robinson, Mark Vanelli, Melissa Nyendak.

**Resources:** Jaspreet Singh.

**Supervision:** Jaspreet Singh, Samantha Robinson, Mark Vanelli, Melissa Nyendak, Meghna Desai.

**Visualization:** Jaspreet Singh, Rajesh Yadav.

**Writing – original draft:** Jaspreet Singh, Rajesh Yadav.

**Writing – review & editing:** Jaspreet Singh, Rajesh Yadav, Samantha Robinson, Mark Vanelli, Melissa Nyendak, Meghna Desai.

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
