## [Decision Letter · Decision Letter 0]

15 Jan 2025

PGPH-D-24-02462

An After-Action Review of COVID-19 Cases and Mitigation Measures at US Mission India, March 2020-July 2021

Dear Dr. Singh Jaspreet,

Thank you for submitting your manuscript to PLOS Global Public Health. After careful consideration, we feel that it has merit but does not fully meet PLOS Global Public Health’s publication criteria as it currently stands. Therefore, we invite you to submit a revised version of the manuscript that addresses the points raised during the review process.

We look forward to receiving your revised manuscript.

Kind regards,

Ekta Saroha

Academic Editor

Journal Requirements:

1. Please provide an Author Summary. This should appear in your manuscript between the Abstract (if applicable) and the Introduction, and should be 150–200 words long. The aim should be to make your findings accessible to a wide audience that includes both scientists and non-scientists. Sample summaries can be found on our website under Submission Guidelines:

https://journals.plos.org/globalpublichealth/s/submission-guidelines#loc-parts-of-a-submission.

2. We have amended your Competing Interest statement to comply with journal style. We kindly ask that you double check the statement and let us know if anything is incorrect.

Additional Editor Comments (if provided):

Reviewers' comments:

Reviewer's Responses to Questions

**Comments to the Author**

1. Does this manuscript meet PLOS Global Public Health’s publication criteria ? Is the manuscript technically sound, and do the data support the conclusions? The manuscript must describe methodologically and ethically rigorous research with conclusions that are appropriately drawn based on the data presented.

Reviewer #1: Yes

Reviewer #2: Yes

2. Has the statistical analysis been performed appropriately and rigorously?

Reviewer #1: Yes

Reviewer #2: Yes

3. Have the authors made all data underlying the findings in their manuscript fully available (please refer to the Data Availability Statement at the start of the manuscript PDF file)?

Reviewer #1: Yes

Reviewer #2: Yes

4. Is the manuscript presented in an intelligible fashion and written in standard English?

Reviewer #1: Yes

Reviewer #2: No

5. Review Comments to the Author

Reviewer #1: I read the article with interest. Thanks for addressing the concerns of the reviewers' in the revised version. The changes made by the authors' are satisfactory.

Reviewer #2: General:

Authors need to correct grammatical errors in the manuscript. For example, in the abstract, there is a sentence, “We described the COVID-19 response across the US Mission India (US Embassy New Delhi, US Consulates – Mumbai, Hyderabad, Chennai and Kolkata) in order use the learnings for a possible future pandemic”. The authors can change it to “to use the learnings”

Abstract:

• Method – instead of “conducted a secondary data analysis”, the authors can consider writing “we analysed the investigation and contact tracing program data”.

• The methods section in the abstract needs more information on data acquisition and analysis reflecting the analysis methods for the results given.

• It is not clear what is meant by “Health unit” – The authors should consider making it clear in the abstract.

• Give the numerator for case fatality rate.

• Does “25% positivity rate” among contacts refer to secondary attack rate? If yes, the authors can use the same term. Despite whether it is SAR or not, the authors need to explain what is this in the methods section.

• Results part is so big with so much data which does not link with the conclusion directly. For example, % of asymptomatic case-patients, number of clusters and the median time between symptoms and notification. The authors can consider giving more information in the methods instead of these information in the results.

• The authors need to make sure that methods, results and conclusions align with one another.

• Keywords to be arranged alphabetically and better to use MeSH words.

Introduction:

• Make sure the objective is given in the past tense

Methods:

• Explain the study site in detail. Give more details which are relevant to understand the interventions and data in the results – what is health unit? What is the administrative structure.

• Give the definition for cases and contacts.

• What tests were used for COVID-19 testing? This should be part of the methods section.

• Who maintained the data on surveillance and contact tracing within each US consulate and how was it consolidated?

• In data analysis – how giving the “frequency of case identification by Mission” can be an epidemic curve? It is not a time distribution.

• Mention what is the source of data for cases in India, Delhi, Chennai, Hyderabad, Kolkata and Mumbai.

Results:

• The control activities are mentioned in detail. However, it is not clear who was responsible for each activity. For example, who handled education sessions? Who did health and safety assessments?

• Vaccination – Who decided what vaccine to be given to whom? Please mention that information.

• Case investigation and contact tracing – what is government-recommended isolation or quarantine. There are four state governments and one federal government involved. Explain which guideline and give references.

• Difference in AR – p-value is given only in the text. It would be better to give in the table as well.

• When there is overall attack rate, why AR by age and gender is not given? Is there a reason? If yes, please mention it.

6. PLOS authors have the option to publish the peer review history of their article (what does this mean? ). If published, this will include your full peer review and any attached files.

**Do you want your identity to be public for this peer review?** For information about this choice, including consent withdrawal, please see our Privacy Policy .

Reviewer #1: No

Reviewer #2: **Yes: ** Manikandanesan Sakthivel

---

## [Decision Letter · Decision Letter 1]

9 Apr 2025

An After-Action Review of COVID-19 Cases and Mitigation Measures at US Mission India, March 2020-July 2021

PGPH-D-24-02462R1

Dear Dr Jaspreet Singh

We are pleased to inform you that your manuscript 'An After-Action Review of COVID-19 Cases and Mitigation Measures at US Mission India, March 2020-July 2021' has been provisionally accepted for publication in PLOS Global Public Health.

Best regards,

Ekta Saroha

Academic Editor

Reviewer Comments (if any, and for reference):

Reviewer's Responses to Questions

**Comments to the Author**

1. If the authors have adequately addressed your comments raised in a previous round of review and you feel that this manuscript is now acceptable for publication, you may indicate that here to bypass the “Comments to the Author” section, enter your conflict of interest statement in the “Confidential to Editor” section, and submit your "Accept" recommendation.

Reviewer #2: All comments have been addressed

2. Does this manuscript meet PLOS Global Public Health’s publication criteria ? Is the manuscript technically sound, and do the data support the conclusions? The manuscript must describe methodologically and ethically rigorous research with conclusions that are appropriately drawn based on the data presented.

Reviewer #2: (No Response)

3. Has the statistical analysis been performed appropriately and rigorously?

Reviewer #2: (No Response)

4. Have the authors made all data underlying the findings in their manuscript fully available (please refer to the Data Availability Statement at the start of the manuscript PDF file)?

Reviewer #2: (No Response)

5. Is the manuscript presented in an intelligible fashion and written in standard English?

Reviewer #2: (No Response)

6. Review Comments to the Author

Reviewer #2: (No Response)

7. PLOS authors have the option to publish the peer review history of their article (what does this mean? ). If published, this will include your full peer review and any attached files.

**Do you want your identity to be public for this peer review?** For information about this choice, including consent withdrawal, please see our Privacy Policy .

Reviewer #2: **Yes: ** Manikandanesan Sakthivel
